# Dissecting *Listeria monocytogenes* Persistent Contamination in a Retail Market Using Whole-Genome Sequencing

Yan Wang,[a] Lijuan Luo,[b] Shunshi Ji,[a] Qun Li,[c] Hong Wang,[c] Zhendong Zhang,[c] Pan Mao,[a] Hui Sun,[a] Lingling Li,[a] Yiqian Wang,[a] Jianguo Xu,[a] Ruiting Lan,[b] Changyun Ye[a]

aState Key Laboratory of Infectious Disease Prevention and Control, National Institute for Communicable Disease Control and Prevention, Collaborative Innovation Center for Diagnosis and Treatment of Infectious Diseases, Chinese Center for Disease Control and Prevention, Beijing, China
bSchool of Biotechnology and Biomolecular Sciences, University of New South Wales, Sydney, Australia
cZigong Center for Disease Control and Prevention, Zigong, Sichuan Province, China

**ABSTRACT** *Listeria monocytogenes* is a foodborne pathogen that can cause invasive disease with high mortality in immunocompromised individuals and can survive in a variety of food-associated environments for a long time. *L. monocytogenes* clonal complex (CC) 87 is composed of ST87 and three other STs and has been identified as the most common subgroup associated with both foods and human clinical infections in China. Therefore, the persistence of CC87 *L. monocytogenes* in food-associated environments poses a significant concern for food safety. In this study, 83 draft genomes of CC87 *L. monocytogenes*, including 60 newly sequenced genomes, were analyzed with all isolates from our previous surveillance in Zigong, Sichuang, China. Sixty-eight of the studied isolates were isolated from one retail market (M1 market), while the others were from seven other markets (M2–M8 markets) in the same city. Whole-genome multilocus sequence typing (wg-MLST) and the whole-genome single nucleotide polymorphism (wg-SNP) analysis were performed. Three persistent contamination routes were identified in the M1 market, caused by 2 clusters (A and B) and a wgST31 type. Cluster A isolates were associated with the persistent contamination in a raw meat stall (M1-S77), while Cluster B isolates caused a persistent contamination in aquatic foods stalls. Five wgST31 isolates caused persistent contamination in a single aquatic stall (M1-S65). A pLM1686-like plasmid was found in all Cluster A isolates. A novel plasmid, pLM1692, a truncated pLM1686 plasmid without the cadmium, and other heavy metal resistance genes were conserved in all wgST31 isolates. By comparing persistent and putative non-persistent isolates, four genes that were all located in the prophage comK might be associated with persistence. These findings enhanced our understanding of the underlying mechanisms of contamination and assist in formulating targeted strategies for the prevention and control of *L. monocytogenes* transmission from the food processing chain to humans.

**IMPORTANCE** Contamination of food by *Listeria monocytogenes* at retail level leads to potential consumption of contaminated food with high risk of human infection. Our previous study found persistent contamination of CC87 *L. monocytogenes* from a retail market in China through pulsed-field gel electrophoresis and multilocus sequence typing. In this study, whole-genome sequencing was used to obtain the highest resolution inference of the source and reasons for persistent contamination; meat grinders and minced meat were the major reservoir of persistent contamination in meat stalls, whereas fish-ponds were the major reservoir in seafood stalls, with different *L. monocytogenes* isolates involved. These isolates carried different properties such as plasmids and prophages, which may have contributed to their ability to survive or adapt to the different environments. Our findings suggest that whole-genome sequencing will be an effective surveillance tool to detect persistent *L. monocytogenes* contamination in retail food markets and to design new control strategies to improve food safety.

Address correspondence to Changyun Ye, yechangyun@icdc.cn, or Ruiting Lan, r.lan@unsw.edu.au.

The authors declare no conflict of interest.

**KEYWORDS** *Listeria monocytogenes*, CC87, CC87-wg-MLST, persistent contamination, whole-genome sequencing, retail market, meat stall, aquatic foods stall

*L*isteria monocytogenes is an opportunistic pathogen responsible for foodborne illness with a variety of manifestations from mild gastroenteritis to severe invasive infections (1). Invasive listeriosis is dangerous for immunocompromised individuals leading to septicemia, meningitis, and encephalitis (2). Consumption of contaminated food, such as meat, fish, dairy, and ready-to-eat food is the major route of human infection (3, 4). *L. monocytogenes* is widespread in a range of environments due to its ability to survive and grow under harsh environmental conditions, such as low pH, low temperatures, and high osmolarity (5). *L. monocytogenes* introduced into a food-associated environment increases the risks of establishing contaminated niches followed by adherence and colonization leading to persistent contamination.

The population of *L. monocytogenes* has been found comprised of four genetic groups with Lineages I to IV, which are further divided by different sequence types (STs) by multilocus sequence typing (MLST), in which closely related STs are classified as a clonal complex (CC) (6). ST87 had been identified as a prevalent sequence type of *L. monocytogenes* of different origins in China (7–9). Together with 3 other STs (namely ST305, ST310, and ST1166), ST87 forms the CC87 clonal complex. CC87 isolates have been rarely reported in Europe and the U.S., except for an outbreak in Spain (10). Recently, subtyping schemes using whole-genome sequences, such as whole-genome multilocus sequence typing (wg-MLST) and whole-genome single nucleotide polymorphism (wg-SNP), have been recognized as effective tools for providing better understandings of the patterns of persistence and dissemination for *L. monocytogenes* within a certain niche or during an outbreak (11–13).

ST87 *L. monocytogenes* is commonly isolated from food products, natural environments, and sporadic listeriosis in China (8, 14, 15). Moreover, this subgroup of *L. monocytogenes* has been gradually recognized as most commonly causing listeriosis in China, with the prevalence rate up to 34% (8, 16, 17). In our previous study we examined the genomic features and the population structure of ST87 *L. monocytogenes* from different regions of China (18). The genomes of ST87 *L. monocytogenes* represent a highly conserved and stable backbone with few accessory genes, which are mainly located in mobile genetic elements, such as prophages and plasmid. A conserved pLM1686-like plasmid with the size of 91 kb was found in a set of ST87 isolates (18). The plasmid carried heavy metal resistance genes, including *cadA1*, *cadA2*, *cadC*, and *copB*, which are related to adaptation to harsh environments (19, 20). We also conducted a retail market surveillance of *L. monocytogenes* for 12 months in 2015, with sampling once a month in Zigong, China (21). Food and environmental samples were collected from eight retail markets. The molecular characteristics and potential persistent contamination of *L. monocytogenes* from meat stalls were identified based on pulsed-field gel electrophoresis (PFGE) and MLST (21). Here, we performed an in-depth study on the CC87 *L. monocytogenes* isolates from the retail market surveillance by using whole-genome sequencing. A panel of 68 CC87 *L. monocytogenes* isolates in a retail market (M1) along with 15 isolates from seven other markets (M2-M8) were selected to (i) confirm the persistent contamination of CC87 *L. monocytogenes* in the retail market, (ii) probe the genetic variations among the persistent CC87 isolates over a year in different niches, (iii) better understand the contamination routes in the retail market, and (iv) explore potential genetic factors of *L. monocytogenes* persistence.

## RESULTS

**Selection of CC87 *Listeria monocytogenes* isolates for whole genome sequencing.** In our previous study of surveillance of retail markets, we identified persistent contamination by the same isolates through PFGE analysis. Here, based on PFGE profiles, we selected part of CC87 *L. monocytogenes* isolates for genomic analysis to further confirm and investigate the persistent contamination that occurred in a retail market (M1

market). Twenty Pulsotype-30 (PT-30) isolates associated with pork products and environments had already been published in Luo et al.'s study (21). Additionally, 43 isolates associated with aquatic products, other meat products (except for pork and aquatic products), and environments with PT-30 from the same market were used in this study. The surveillance of the aquatic products, other meat products, and environments was done at the same time as the study of Luo et al., and isolates obtained were typed similarly. Except for the isolates from pork stalls, there were 78 isolates obtained in the M1 market over the 12-month survey. By serotype, 57 were 1/2b, by ST, 13 ST87 and 40 ST1166 (both STs belonging to CC87). By PFGE, 50 isolates were PT30, 43 of which were selected for this study. The distribution of the isolates among the aquatic stalls and other meat stalls over the sampling months are presented in Fig. S1 in the supplemental material. For comparison, we chose two groups of isolates from the same surveillance: (i) 9 PT30 *L. monocytogenes* isolates from other markets (M2–M8), and (ii) 11 non-PT30 (PT27, PT315, PT317, PT331, PT322, and PT324) but CC87 isolates from all markets. The background information of each *L. monocytogenes* isolate is listed in Table 1. All the non-M1 markets were randomly named M2 to M8. Sixty isolates were subjected to sequencing in this study, and the other 23 isolates had been sequenced in our previous study (18). The complete genome of isolate ICDC-LM188 (accession no. CP015593), belonging to CC87/ST87, was used as a reference.

**Whole-genome MLST analysis on selected *L. monocytogenes* isolates.** Based on the classic seven-locus MLST scheme (6), our *in silico* analysis of the genomes divided the 83 isolates into two sequence types, with 48 ST87 isolates and 35 ST1166 isolates. The two STs belonged to CC87 and were separated by the *ldh* locus with one SNP difference. To provide further insight into the relationship and/or changes among these CC87 *L. monocytogenes* isolates, we performed whole-genome MLST (wgMLST) analysis of CC87 isolates, referred to as CC87-wg-MLST here to type all isolates. Note that we did not attempt to develop a CC87 wgMLST scheme and were only using this approach to obtain the best resolution for typing of the isolates rather than using the species-specific core genome MLST scheme (22). The gene-by-gene comparison was performed using the FAST-Gep program (23). A total of 2,723 shared loci and 378 polymorphic alleles were identified. The 83 isolates were typed into 53 wg-MLST types (wgSTs). The isolates were previously typed by PFGE, 86.7% of which belonged to the same PFGE type (PT30). CC87-wg-MLST showed a higher level of discrimination than PFGE. The 37 ST87-PT30 *L. monocytogenes* isolates were further divided into 24 wgSTs with three predominant types including wgST1 ($n = 5$), wgST2 ($n = 5$), and wgST31 ($n = 5$), while the 35 ST1166-PT30 isolates were further divided into 18 wgSTs with one predominant type, wgST14 ($n = 19$). All non-PT30 CC87 *L. monocytogenes* isolates were separated by CC87-wg-MLST with their specific types. Based on the CC87-wg-MLST allelic profiles, the minimum spanning tree was constructed with two clusters and 21 unclustered wgSTs identified (Fig. 1). Clusters were defined based on a cutoff of 4 alleles using the Silhouette index. The two clusters were named Cluster A (22 isolates) and Cluster B (35 isolates), with four alleles as the maximum pairwise allelic differences within each cluster. All except one unclustered wgST contained one isolate only. Unclustered wgST31 contained five isolates.

**Cluster A isolates caused persistent and cross contamination in raw meat stalls.** All Cluster A *L. monocytogenes* isolates were isolated from the same retail market M1 during February to July and October 2015 (Table 1), with the exception of one isolate, ICDC-LM1201, being isolated from retail market M2 in November 2014. More than half (12/22) of Cluster A isolates were repeatedly isolated from the same stall (M1-S77) over a 6-month period (February to July, 2015) in a monthly sampling (Fig. 2). Moreover, 6 of these 12 isolates were detected in the meat grinder of the M1-S77 stall, which were separately isolated in a 6-month period. Additionally, five Cluster A isolates were separately isolated from five non-M1-S77 stalls in the M1 market in May 2015 (Fig. 2).

By CC87-wg-MLST, Cluster A isolates (all ST87-PT30) were divided into 12 types. wgST1 and wgST2 were the predominant types, which were linked with wgST3 by one allelic difference (Fig. 1). Isolate ICDC-LM1637, which had a different PFGE type (PT319), was identified

**TABLE 1** *Listeria monocytogenes* isolates included in this study

| Isolate name | Market and stall | Sampling mo and yr | PFGE profile | Sequence type | wg-MLST | Cluster |
|---|---|---|---|---|---|---|
| ICDC-LM1253 | M1-S77 | Feb, 2015 | 30 | 87 | 1 | Cluster A |
| ICDC-LM1341 | M1-S77 | Mar, 2015 | 30 | 87 | 1 | Cluster A |
| ICDC-LM1457 | M1-S77 | Apr, 2015 | 30 | 87 | 1 | Cluster A |
| ICDC-LM1496 | M1-S23 | May, 2015 | 30 | 87 | 1 | Cluster A |
| ICDC-LM1605 | M1-S77 | Jun, 2015 | 30 | 87 | 1 | Cluster A |
| ICDC-LM1201 | M2-S19 | Nov, 2014 | 30 | 87 | 2 | Cluster A |
| ICDC-LM1346 | M1-S77 | Mar, 2015 | 30 | 87 | 2 | Cluster A |
| ICDC-LM1459 | M1-S77 | Apr, 2015 | 30 | 87 | 2 | Cluster A |
| ICDC-LM1515 | M1-S13 | May, 2015 | 30 | 87 | 2 | Cluster A |
| ICDC-LM1542 | M1-S22 | May, 2015 | 30 | 87 | 2 | Cluster A |
| ICDC-LM1249 | M1-S77 | Feb, 2015 | 30 | 87 | 3 | Cluster A |
| ICDC-LM1509 | M1-S77 | May, 2015 | 30 | 87 | 3 | Cluster A |
| ICDC-LM1508 | M1-S77 | May, 2015 | 30 | 87 | 4 | Cluster A |
| ICDC-LM1604 | M1-S77 | Jun, 2015 | 30 | 87 | 5 | Cluster A |
| ICDC-LM1250 | M1-S77 | Feb, 2015 | 30 | 87 | 6 | Cluster A |
| ICDC-LM1523 | M1-S58 | May, 2015 | 30 | 87 | 7 | Cluster A |
| ICDC-LM1452 | M1-S58 | Apr, 2015 | 30 | 87 | 8 | Cluster A |
| ICDC-LM1449 | M1-S14 | Apr, 2015 | 30 | 87 | 9 | Cluster A |
| ICDC-LM1404 | M1-S58 | Mar, 2015 | 30 | 87 | 10 | Cluster A |
| ICDC-LM1637 | M1-S77 | Jul, 2015 | 319 | 87 | 11 | Cluster A |
| ICDC-LM1514 | M1-S14 | May, 2015 | 30 | 87 | 12 | Cluster A |
| ICDC-LM1728 | M1-S45 | Oct, 2015 | 30 | 87 | 13 | Cluster A |
| ICDC-LM1208 | M1-S64 | Jan, 2015 | 30 | 1166 | 14 | Cluster B |
| ICDC-LM1234 | M1-S60 | Jan, 2015 | 30 | 1166 | 14 | Cluster B |
| ICDC-LM1255 | M1-S63 | Feb, 2015 | 30 | 1166 | 14 | Cluster B |
| ICDC-LM1378 | M1-S61 | Mar, 2015 | 30 | 1166 | 14 | Cluster B |
| ICDC-LM1453 | M1-S63 | Apr, 2015 | 30 | 1166 | 14 | Cluster B |
| ICDC-LM1473 | M1-S61 | Apr, 2015 | 30 | 1166 | 14 | Cluster B |
| ICDC-LM1525 | M1-S61 | May, 2015 | 30 | 1166 | 14 | Cluster B |
| ICDC-LM1598 | M1-S61 | Jun, 2015 | 30 | 1166 | 14 | Cluster B |
| ICDC-LM1599 | M1-S60 | Jun, 2015 | 30 | 1166 | 14 | Cluster B |
| ICDC-LM1602 | M1-S63 | Jun, 2015 | 30 | 1166 | 14 | Cluster B |
| ICDC-LM1603 | M1-S62 | Jun, 2015 | 30 | 1166 | 14 | Cluster B |
| ICDC-LM1621 | M1-S62 | Jul, 2015 | 30 | 1166 | 14 | Cluster B |
| ICDC-LM1626 | M1-S65 | Jul, 2015 | 30 | 1166 | 14 | Cluster B |
| ICDC-LM1638 | M1-S63 | Jul, 2015 | 30 | 1166 | 14 | Cluster B |
| ICDC-LM1744 | M1-S62 | Oct, 2015 | 30 | 1166 | 14 | Cluster B |
| ICDC-LM1798 | M1-S63 | Oct, 2015 | 30 | 1166 | 14 | Cluster B |
| ICDC-LM1807 | M1-S51 | Nov, 2015 | 30 | 1166 | 14 | Cluster B |
| ICDC-LM1821 | M1-S60 | Dec, 2015 | 30 | 1166 | 14 | Cluster B |
| ICDC-LM1844 | M1-S61 | Jan, 2015 | 30 | 1166 | 14 | Cluster B |
| ICDC-LM1207 | M1-S63 | Jan, 2015 | 30 | 1166 | 15 | Cluster B |
| ICDC-LM1845 | M1-S60 | Jan, 2015 | 30 | 1166 | 16 | Cluster B |
| ICDC-LM1405 | M1-S62 | Mar, 2015 | 30 | 1166 | 17 | Cluster B |
| ICDC-LM1500 | M1-S62 | May, 2015 | 30 | 1166 | 18 | Cluster B |
| ICDC-LM1507 | M1-S63 | May, 2015 | 30 | 1166 | 19 | Cluster B |
| ICDC-LM1524 | M1-S60 | May, 2015 | 30 | 1166 | 20 | Cluster B |
| ICDC-LM1627 | M1-S60 | Jul, 2015 | 30 | 1166 | 21 | Cluster B |
| ICDC-LM1658 | M1-S62 | Aug, 2015 | 30 | 1166 | 22 | Cluster B |
| ICDC-LM1661 | M1-S63 | Aug, 2015 | 30 | 1166 | 23 | Cluster B |
| ICDC-LM1662 | M1-S60 | Aug, 2015 | 30 | 1166 | 24 | Cluster B |
| ICDC-LM1746 | M1-S60 | Oct, 2015 | 30 | 1166 | 25 | Cluster B |
| ICDC-LM1812 | M1-S61 | Nov, 2015 | 30 | 1166 | 26 | Cluster B |
| ICDC-LM1822 | M1-S61 | Dec, 2015 | 30 | 1166 | 27 | Cluster B |
| ICDC-LM1540 | M1-S61 | May, 2015 | 30 | 1166 | 28 | Cluster B |
| ICDC-LM1342 | M1-S63 | Mar, 2015 | 30 | 1166 | 29 | Cluster B |
| ICDC-LM1420 | M1-S62 | Apr, 2015 | 30 | 1166 | 30 | Cluster B |
| ICDC-LM1692 | M1-S65 | Sep, 2015 | 30 | 87 | 31 | Cluster C |
| ICDC-LM1693 | M1-S65 | Sep, 2015 | 30 | 87 | 31 | Cluster C |
| ICDC-LM1713 | M1-S65 | Sep, 2015 | 30 | 87 | 31 | Cluster C |
| ICDC-LM1782 | M1-S65 | Dec, 2015 | 30 | 87 | 31 | Cluster C |
| ICDC-LM1784 | M1-S65 | Dec, 2015 | 30 | 87 | 31 | Cluster C |

**TABLE 1** (Continued)

| Isolate name | Market and stall | Sampling mo and yr | PFGE profile | Sequence type | wg-MLST | Cluster |
|---|---|---|---|---|---|---|
| ICDC-LM1233 | M1-S42 | Jan, 2015 | 30 | 87 | 32 | nonclustered |
| ICDC-LM1203 | M4-S51 | Jan, 2015 | 331 | 87 | 33 | nonclustered |
| ICDC-LM1497 | M8-S14 | May, 2015 | 30 | 87 | 34 | nonclustered |
| ICDC-LM1502 | M5-S33 | May, 2015 | 30 | 87 | 35 | nonclustered |
| ICDC-LM1218 | M6-S15 | Jan, 2015 | 30 | 87 | 36 | nonclustered |
| ICDC-LM1620 | M1-S67 | Jul, 2015 | 345 | 87 | 37 | nonclustered |
| ICDC-LM1625 | M1-S65 | Jul, 2015 | 30 | 87 | 38 | nonclustered |
| ICDC-LM1659 | M1-S65 | Aug, 2015 | 27 | 87 | 39 | nonclustered |
| ICDC-LM1520 | M3-S99 | May, 2015 | 317 | 87 | 40 | nonclustered |
| ICDC-LM1175 | M2-S5 | Nov, 2014 | 30 | 87 | 41 | nonclustered |
| ICDC-LM1197 | M2-S15 | Nov, 2014 | 30 | 87 | 42 | nonclustered |
| ICDC-LM1204 | M4-S77 | Jan, 2015 | 315 | 87 | 43 | nonclustered |
| ICDC-LM1373 | M1-S51 | Mar, 2015 | 331 | 87 | 44 | nonclustered |
| ICDC-LM1361 | M2-S80 | Mar, 2015 | 30 | 87 | 45 | nonclustered |
| ICDC-LM1296 | M1-S58 | Feb, 2015 | 323 | 87 | 46 | nonclustered |
| ICDC-LM1220 | M4-S102 | Jan, 2015 | 324 | 87 | 47 | nonclustered |
| ICDC-LM1492 | M6-S28 | May, 2015 | 30 | 87 | 48 | nonclustered |
| ICDC-LM1572 | M5-S13 | Jun, 2015 | 322 | 87 | 49 | nonclustered |
| ICDC-LM1665 | M1-S42 | Aug, 2015 | 30 | 87 | 50 | nonclustered |
| ICDC-LM1848 | M2-S83 | Jan, 2015 | 30 | 87 | 51 | nonclustered |
| ICDC-LM1674 | M7-S16 | Aug, 2015 | 27 | 87 | 52 | nonclustered |

as wgST11 and was isolated from stall M1-S77 in July 2015, which was the last sampling time point when *L. monocytogenes* was positive in this meat stall.

Among the Cluster A isolates, 13 out of 2,723 core genes were polymorphic based on CC87-wg-MLST (Table S2). In the 13 polymorphic genes, 10 were caused by SNPs, one caused by 1-bp in/del, and two caused by carrying different numbers of tandem repeats. An assembly-free whole genome SNP-calling strategy was further used to

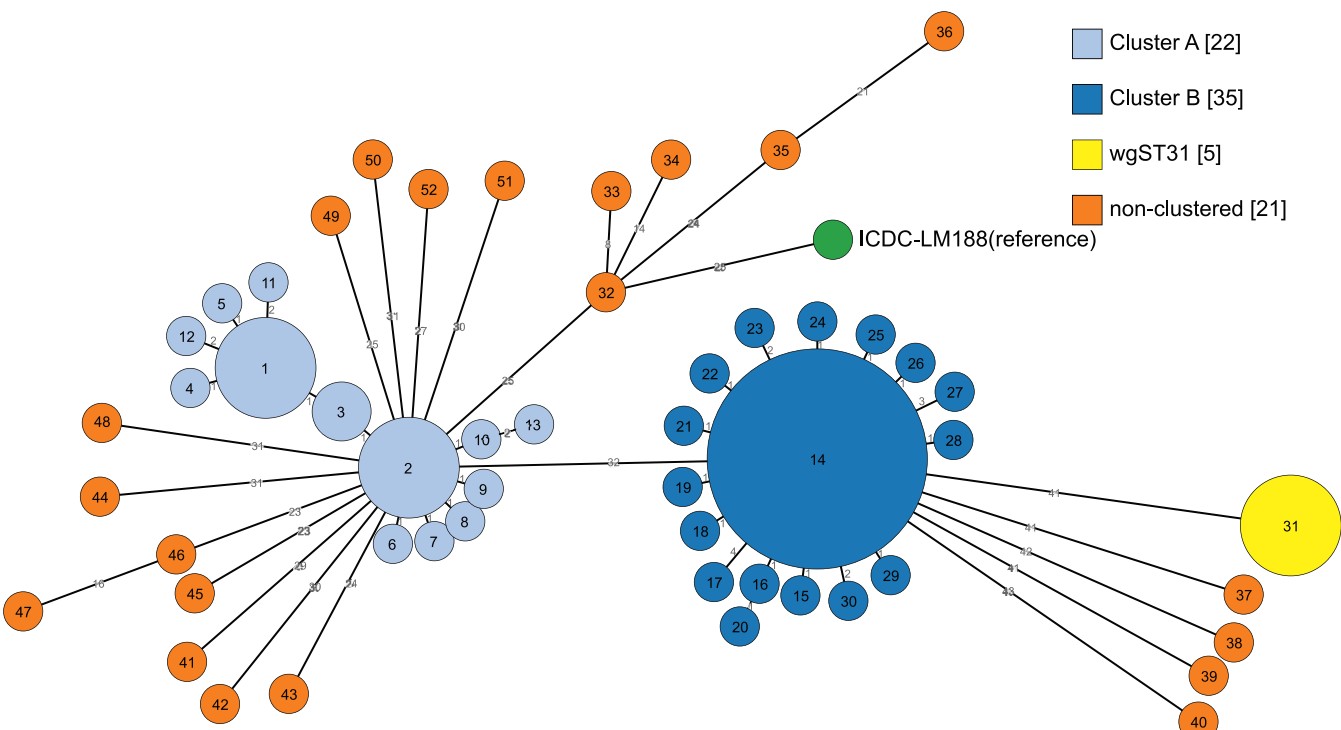

**FIG 1** Minimum spanning trees based on CC87-wg-MLST allelic profiles present the relationship among the 83 CC87 *L. monocytogenes* isolates along with the reference strain ICDC-LM188. The number in each circle is wg-MLST sequence type (wgST), and the size of each circle is proportional to the number of isolates. The number of allelic differences between each wgST was labeled on the connecting line between two circles. Cluster A and Cluster B, which were two major groups of persistent isolates identified and are marked in light blue and dark blue, respectively. wgST31 is a unique single ST that contains more than one isolate and is marked in yellow. All the putative nonpersistent isolates are marked in orange. The reference strain ICDC-LM188 is marked in green.

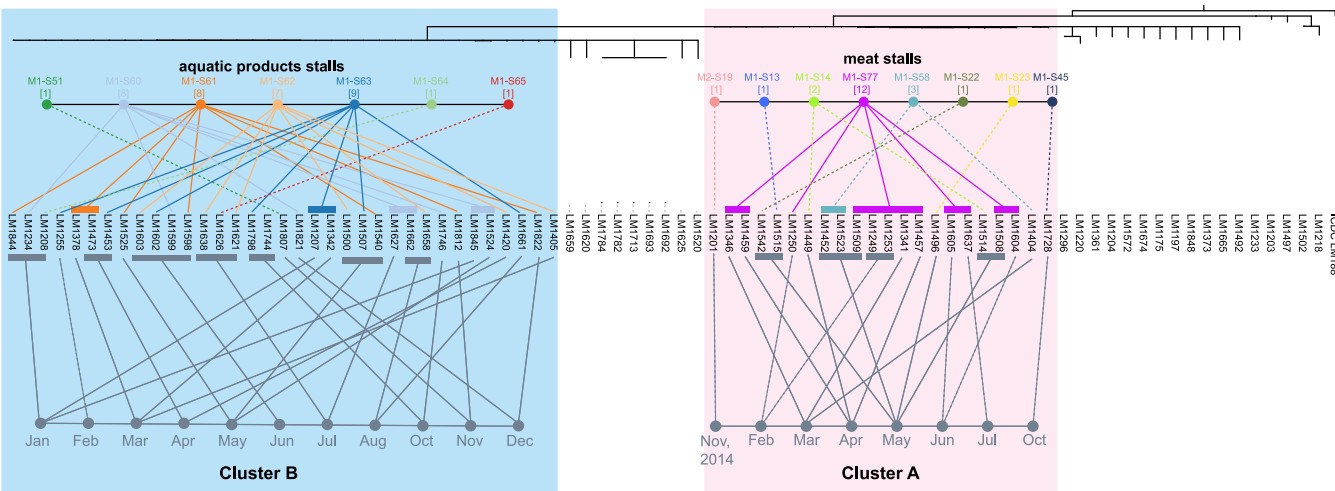

**FIG 2** The dendrogram of hierarchical clustering of isolates based on CC87-wg-MLST allelic profiles. The metadata of Cluster A (in light pink shadow) and Cluster B (in light blue shadow) isolates, including isolating time and isolating stall, are mapped to the corresponding isolate.

identify any extra SNPs outside the CC87-wgMLST genes and intergenic regions among the Cluster A isolates. A total of 12 SNPs were found with 10 being the same SNPs as identified by CC87-wg-MLST. Of these SNPs, eight were non-synonymous, two were synonymous, and two were in the intergenic regions. All the amino acid substitutions caused by nonsynonymous SNPs were predicted to be non-deleterious using software SIFT (24).

Pangenome analysis of Cluster A isolates using Roary predicted 2,952 hard-core genes (found in ≥99% of genomes), two soft-core genes (found in 95–99% of genomes), seven shell genes (found in 15–95% of genomes), and 70 cloud genes (found in <15% of genomes). Cluster A core-genome (2,952 genes) is 8.4% larger than the CC87 core-genome (2,723 genes). Interestingly, 66 of the 70 cloud genes were contiguous and belonged to a prophage, $\varphi$tRNA-Arg, which was inserted into the downstream of tRNA-Arg in three Cluster A isolates, including ICDC-LM1201, ICDC-LM1523, and ICDC-LM1604. There were no SNP differences in the prophages $\varphi$tRNA-Arg between ICDC-LM1201 and LM1523, both of which had two SNP differences from the isolate ICDC-LM1604. In addition, all Cluster A isolates harbored a 91 kb plasmid that was reported to be carried by a subset of ST87 isolates in our previous study (18).

**Cluster B isolates associated with persistence contamination in aquatic food products stalls.** The majority (91.4%) of Cluster B isolates were isolated from four neighboring aquatic products stalls (M1-S60 to M1-S63) in the M1 market, and the remaining four Cluster B isolates were isolated from three stalls, M1-S51(in November 2015), M1-S64 (in January 2015), and M1-S65 (in July and December 2015), which were located on the periphery of M1-S60 to M1-S63. All Cluster B isolates were ST1166-PT30.

By CC87-wg-MLST, 35 Cluster B isolates were divided into 17 wgMLST types, wgST14 contained 19 isolates while the other wgSTs had a single isolate. Among the 16 minor wgSTs, 11 wgSTs had one allelic difference from wgST14 (Fig. 1). Twenty polymorphic genes were identified and the gene functions tend to be related to metabolisms, such as the metabolism and transport of carbohydrates and inorganic ions (Table S3). Notably, two polymorphic genes encoded cell surface proteins were shared with the ones identified in Cluster A isolates.

By assembly-free wgSNP analysis, a total of 22 SNPs were identified among the 35 Cluster B isolates. Four SNPs were located in intergenic regions, while 18 SNPs were located in the coding genes, including 13 non-synonymous SNPs and five synonymous SNPs. One nonsynonymous SNP was located in the gene encoding a phage tail tape measure protein (ICDC-LM188 locus tag: A6K41_11830), which was not included in the

alleles of the CC87-wg-MLST. All the amino acids substitutions caused by nonsynonymous SNPs were predicted to be non-deleterious using SIFT. In isolate ICDC-LM1420, one SNP resulted in a premature stop codon in the gene coding for PTS sorbose transporter subunit IIC (A6K41_00130 in ICDC-LM188). Pangenome analysis using Roary predicted 2906 hard core genes, 7 soft core genes, 7 shell genes and 5 cloud genes. All the Cluster B isolates harbored a prophage, which was inserted downstream the tRNA-Arg. The prophage showed no variation among the Cluster B isolates except for one isolate, ICDC-LM1525, which failed to be fully assembled as one contig. This prophage showed 92% identity and 84% coverage with the prophage $\varphi$tRNA-Arg from a Cluster A isolate ICDC-LM1201. No plasmid was found in any of the Cluster B isolates.

**Characterizations of the five wgST31 isolates from aquatic food products stall.** Five isolates isolated from aquatic samples in the stall M1-S65 in September 2015 ($n = 3$) and December 2015 ($n = 2$) shared the same CC87-wg-MLST type (wgST31). Further assembly free wgSNP analysis found one SNP in an intergenic region and another synonymous SNP of a stop codon (locus tag in ICDC_LM188: A6K41_11850). The three isolates isolated in September 2015 showed different SNP profiles while the two isolates isolated in December 2015 shared the same SNP profile with one of the isolates in September. There was no prophage inserted into the end of tRNA-Arg in the wgST31 isolates. However, all five wgST31 isolates shared another prophage that was inserted between the FosX/FosE/FosI family fosfomycin resistance gene (locus tag in ICDC_LM188: A6K41_08870) and 23S rRNA (uracil-5-)-methyltransferase RumA gene (locus tag in ICDC_LM188: A6K41_08875), and was named $\varphi$rumA here.

A novel plasmid, named pLM1692 was identified in all five wgST31 isolates. pLM1692 is about 60 kb in size and encodes 65 genes. Fifty-seven of the genes showed a high level of similarity with genes in a larger plasmid, pLM1686, as previously reported (Table S4) (19). In comparison with pLM1686, pLM1692 had a deletion of 30 kb which included a transposase gene, heavy metal resistance genes (*cadAC*), multi-copper oxidase gene (*mco*), and copper transporter gene (*copB*).

**A new prophage $\varphi$comK inserted into *comK* and its association with persistent isolates.** Clusters A and B, as well as wgST31 isolates, were considered as persistent isolates while the remaining 21 isolates were considered as non-persistent isolates based on whether they were isolated multiple times or only once during the surveillance. The latter isolates had their unique wg-MLST profiles (from wgST32 to wgST52) with at least more than four allelic differences with each other (Fig. 1). These isolates all belonged to ST87 and 11 of the 21 isolates belonged to PT30.

Scoary was used to determine any associations between the accessory genome and the trait of persistence. With a naive *P* value of <0.05, there were no genes exclusively present in the genomes from all the persistent isolates. No genes were identified to be associated with persistence with 100% sensitivity and specificity. However, four genes located in a prophage ($\varphi$comK) were identified in the genomes of all the persistent isolates and some of the non-persistent isolates with 100% sensitivity and 41.67%~45.83% specificity, two of the genes were homologous to *lmo2321* and *lmo2322* (encoding gp44 [bacteriophage A118]) of the strain EGD-e with an amino acid sequence identity of 81% and 76% respectively, and the other two genes were annotated as hypothetical proteins.

We further characterized the $\varphi$comK in our isolates. Eighty-one of the 83 CC87 isolates of *L. monocytogenes* harbored the prophage $\varphi$comK, except for two putative non-persistent isolates. The complete sequences of $\varphi$comK were available in 65 isolates, of which 58 isolates were persistent isolates and seven were putative non-persistent isolates. Seven types of $\varphi$comK were identified and were named as $\varphi$comK-type-1 to $\varphi$comK-type-7. The $\varphi$comK-type-1 prophages with the size of 40,671 bp were observed in all Cluster A isolates with 100% identity to each other. $\varphi$comK-type-2 prophages were observed in the isolates in Cluster B and wg-MLST profile type-3**1**. Interestingly, it was also found in a putative non-persistent isolate ICDC-LM1218, which was isolated from the market M6. $\varphi$comK-type-2 has a size of 38,304 bp. Two variants of $\varphi$comK-type-2 were found with 1-bp deletion and the other with one SNP. These two variants of $\varphi$comK were both carried by Cluster B isolates (ICDC-LM1207 and

ICDC-LM1405). The $\varphi$comK-type-3 prophage with the size of 40,823 bp was harbored by two putative non-persistent isolates, ICDC-LM1203 and ICDC-LM1233, which were isolated from M4 and M1 markets respectively. The $\varphi$comK_type-5 to type 7 prophages were each harbored by one non-persistent isolate.

To investigate the diversity of these seven types of $\varphi$comK, the prophage sequences in each type were chosen for pan-genome analysis using Roary. Only one gene was identified as core gene ($\geq$ 99% of the genomes), 137 genes were shell genes (15-95% of the genomes), and 64 cloud genes (<15% of the genomes). The Type 5 prophage was found to be the most diverse, sharing only one gene with other types of prophages (Fig. S2). In addition, 70 and 58 coding genes were predicted in $\varphi$comK_type-1 and $\varphi$comK_type-2 prophages, respectively, among which 24 genes that were annotated as hypothetical proteins were shared by both types of $\varphi$comK.

## DISCUSSION

The repeated introductions and the persistent survival of *L. monocytogenes* led to the contamination of food and food-associated environments, posing severe challenges to food safety. *L. monocytogenes* is a foodborne pathogen and should be closely monitored to prevent contamination in food production and distribution processes. Timely detection of contaminants and tracing the contamination routes are of great significance to the assurance of food safety. In the last decade, analyses of genome-scale data have been used to trace outbreaks-associated and persistent *L. monocytogenes* isolates with higher discriminatory power than conventional molecular subtyping methods, such as PFGE and MLST (11, 13). Whole-genome-based analysis can provide insight into more refined relationships and the dynamics of microevolution among a set of isolates in a certain niche. Further, it also can provide more genetic information on molecular determinants that are attributions of adaptation to stress and/or enhanced potentials (25). Several studies have focused on the characterization of persistent isolates over a long-term time frame with relative long-time intervals (13, 26). In this study, we studied a subset of CC87 *L. monocytogenes* isolates with the same PFGE patterns that were obtained from a 12-month surveillance project of retail food markets (21, 27), to further shed light on the persistent contamination, based on whole genome sequences, and identified two major persistent contaminations across the market. Whole-genome sequencing offered much higher resolution than PFGE as most isolates were the same PFGE type and were divided into two major clusters, allowing identification of contamination patterns and persistence of isolates.

**Persistent CC87 *L. monocytogenes* contamination events occurring in the studied market.** Whole-genome sequencing identified one persistent contamination event caused by Cluster A in the meat market. Genomic diversity was observed within Cluster A with 13 wgMLST types. All cluster A isolates were isolated from the same meat market M1 except for one isolate (ICDC-LM1201) from another market (M2) a few months earlier in November 2014. ICDC-LM1201 from market M2 was wgMLST Type-2, which was first detected in March 2015 in market M1, and three wgMLST types (Type-1, Type-3 and Type-6) were detected in February 2015 in market M1. Thus, Cluster A must have been circulating in different markets or upstream of a common source well before their first appearance in the markets. Interestingly, Cluster A was only detected once (based on PFGE type) in market M2 (21), but the reason of not persisting in M2 was unknown.

Stall M1-S77 was previously found to be repeatedly contaminated with the same PFGE type (21). Whole-genome sequence analysis provided a more detailed picture. For the 12 isolates sequenced, seven wgMLST types were found. These isolates were obtained from February 2015 to July 2015. It is possible that some of the genomic diversity was developed within the stall as wgST3, 4, 6, and 11 were found in that stall only. Stall M1-S77 had a meat grinder that was mainly used to grind meat from this stall and occasionally from other stalls. It was also likely that meat contaminated with Cluster A or other types entered the grinder and became a reservoir of continuous contamination as cleaning of the meat grinder may have been ineffective. There were unique wgSTs isolated in other stalls, that were not found in M1-S77, suggesting that

the contamination of Cluster A in market M1 was widespread and was likely persisting in the entire meat market. Contamination of M1-S77 was likely to have been exacerbated by contamination of the meat grinder. It would be interesting to do further sampling to determine whether Cluster A remained in the market without intervention in the past few years.

The second persistent contamination event was caused by Cluster B isolates and had occurred in aquatic products and their related environments also in M1 market. Aquatic stalls selling live fish were in a different building level from the meat stalls in the same market. The contamination was concentrated around four stalls (M1-S60 to M1-S63) of aquatic products through the 12-month sampling period. Cluster B had only one predominant wgST (wgST14) along with 18 minority wgSTs. wgST14 was isolated over the 12-month sampling period, suggesting that it was well established in the stall environment. Since each stall had a fishpond to keep the fishes alive, it was highly likely Cluster B had established in the fishponds as a reservoir. Based on this observation, a real-time whole-genome sequencing monitoring for targeted cleaning and disinfection of the aquatic product stalls and fishponds, in particular in M1-S60 to M1-S63, could reduce or eliminate this persistent contamination.

There were five wgST31 isolates obtained in September and December 2015 from the M1-S65 stall, where one Cluster B isolate ICDC-LM1626 was isolated earlier in July. The latter differed from the wgST31 isolates by 41 alleles, suggesting two different contamination sources were independently introduced into M1-S65. wgST31 remained in the stall at least until the end of the year, while Cluster B was not recovered in this stall late in the year. We were unable to infer the causes of persistence of wgST31 in this stall.

**Mobile genetic elements (MGEs) and possible roles in the persistence and environmental adaptation of CC87 *L. monocytogenes* isolates.** ST87 *L. monocytogenes* has a highly stable genome backbone along with different prophages as major components of its accessory genome (18). MGEs may be involved in the adaptation and persistence of certain persistent *L. monocytogenes* clones. In this study, we found two new prophages ($\varphi$comK type-1 and $\varphi$comK type-2) in the persistent isolates. Cluster A isolates harbored Type-1 $\varphi$comK, and Cluster B isolates harbored two prophages, $\varphi$tRNA-Arg and Type-2 $\varphi$comK, while wgMLST Type-31 isolates harbored two prophages, $\varphi$rumA and Type-2 $\varphi$comK. Additionally, three Cluster A isolates, ICDC-LM1201, ICDC-LM1523, and ICDC-LM1604, harbored an additional prophage $\varphi$tRNA-Arg, which was different from the prophage $\varphi$tRNA-Arg in Cluster B isolates. These three isolates were of different wgSTs, suggesting that they acquired $\varphi$tRNA-Arg prophages independently.

Moreover, scoary analysis identified four phage genes that were possibly associated with persistence. However, the functions of the genes were unknown and thus it is unknown whether they play any role in adaptation. The *comK* site was previously found to be a phage insertion site hot spot with different phages inserted at the site (28). The *comK* site has been found to play a regulatory role in niche adaptation and virulence (28–30). Insertion of a phage disrupts the *comK* gene, while excision of the prophage reverts *comK* to an intact and functional gene which encodes the master activator of the DNA uptake competence (Com) system (29). In this study, we found seven unrelated $\varphi$comK phages at the *comK* site. The insertion of multiple phages in the *comK* site may suggests that the Com system was quite frequently switched on and off through phage excision and insertion. The new phages may carry genes for adaptation to different environments.

Our previous study had reported that a 91-kb plasmid pLM1686 was carried by a subset of ST87 *L. monocytogenes* (18, 19). In this study, we found that the plasmid pLM1686 was exclusively carried by Cluster A isolates. The cadmium and some other heavy metal resistance genes might be necessary for the adaptation of Cluster A isolates in the meat products environments. Carriage of pLM1686 has been shown to enhance growth and biofilm formation, salinity tolerance, cell invasion, and cytotoxicity (20). Similar plasmids were also found in other STs and were found to contribute to

tolerance against elevated temperature, salinity, acidic environments, oxidative stress, and disinfectants (31). The finding of pLM1686 carrying heavy metal resistance genes in persistent *L. monocytogenes* isolates suggests that the plasmid contributes to their persistence in the meat market environment. However, a smaller plasmid, 60-kb pLM1692, which was a truncated pLM1686 that has lost the cadmium, some other heavy metal resistance genes, and related transposable elements, was carried by the five wgST31 isolates isolated from the aquatic food or environments.

**Recommendations for control and prevention of *L. monocytogenes* contamination in retail markets.** The contamination of *L. monocytogenes* depends not only on the biological characteristics of bacterium itself, such as colonization ability, biofilm formation ability, and disinfectant resistance, but also on the living environments of the bacterium. The persistent contamination caused by Cluster A in this study was to a certain extent attributed to the hard-to-clean structure of the meat grinder. A combination of the repeated introduction of new strains and continuous survival in the grinder may have led to the persistent and cross contamination in meat stalls in the M1 market. Cluster B isolates were most likely surviving in the aquatic environments, which were live fish fishponds. Inadequate cleaning and disinfection of the fishponds were likely the main reason for the persistent survival of Cluster B isolates. The contrasting findings of two different *L. monocytogenes* clones surviving in two different environments underscore the importance of genomic analysis of the contaminating strains to understand the underlying mechanisms of contamination and targeted strategies for the prevention and control of *L. monocytogenes* transmission from the food processing chain to humans.

**Conclusions.** In summary, we identified three persistent subtypes of CC87 *L. monocytogenes*: (i) Cluster A isolates that persistently survived in a stall with a meat grinder, and possibly caused cross-contamination with other meat stalls in April and May 2015; (ii) cluster B isolates that persistently contaminated four aquatic food stalls throughout the year; and (iii) wg-MLST-Type-31 isolates that exclusively contaminated one aquatic food stall. Although no clear genetic determinants of persistence were identified, the mobile genetic elements, including prophages and plasmids, were identified to be cluster-specific, likely contributing to different fitness and adaptation in the complex ecosystems. Our study showed that the application of whole-genome sequence analysis can significantly inform food safety surveillance of *L. monocytogenes* and control efforts in retail markets.

## MATERIALS AND METHODS

**DNA Extraction, whole-genome sequencing, assembly, and annotation.** DNA extraction was performed using the Wizard Genomic DNA purification kit (Promega corporation, 2800 Woods Hollow Road, Madison, WI 53711 USA), according to the manufacturer's instructions for Gram-positive bacteria. Total DNA obtained was subjected to quality control by agarose gel electrophoresis and quantified by Qubit. A pair-end library with an insert size of 500 bp was constructed and sequenced using an Illumina HiSeq X by PE150 strategy at the Beijing Novogene Bioinformatics Technology Co, Ltd. After removing the low-quality reads and adapter reads, the filtered reads were assembled by SOAPdenovo v.1.05 to generate scaffolds, and then annotation was performed using the NCBI Prokaryotic Genomes Automatic Annotation Pipeline (PGAAP).

**Whole genome analysis.** The seven-locus MLST was performed *in silico* using the BIGSdb-Lm database (https://bigsdb.pasteur.fr/listeria/). A genome-by-genome allele calling program without a containing scheme, named CC87-wg-MLST here, was performed using the Fast-Genome Profile (Fast-GeP), which is freely available from https://github.com/jizhang-nz/fast-GeP (23). The minimum spanning trees were constructed using the software GrapeTree v.1.5.0 with the MSTreeV2 method to present the relationship among the isolates based on whole genomic scale allelic profiles (32). The different sequences of all the shared loci among the isolates were assigned different alleles, and the combination of the alleles of the shared loci defined the whole genome sequence type (wgST). The core/pan-genome analyses were performed using Roary v.3.13.0 (33) with default parameters (the minimum percentage identity for blastp set to 95, and percentage of isolates a gene must be in to be core set to 99). The general feature format 3 (gff3) files generated by Prokka v1.14.4 (34) were used as input data for Roary. According to the harboring rate of each gene, all the pan genes were classified as hard-core genes (in ≥99% of genomes), soft-core genes (in 95–99% of genomes), shell genes (15–95% of genomes), and cloud genes (in <15% of genomes). Core genome SNPs calling was performed using Snippy v.4.4.5 (https://github.com/tseemann/snippy).

**Analysis of prophages and plasmids.** Prophages were identified using the online webserver PHASTER (http://phaster.ca) and were compared by BLASTN among each type of prophages. To investigate the diversity of prophage $\varphi$comK in CC87 isolates, Roary was used with the options of -i (minimum percentage identity for blastp) and -cd (percentage of isolates a gene must be in to be core) of 96 and 99 respectively. The plasmids in the isolates were identified by BLASTN searches using the sequence of pLM1686 as a query against each assembled sequence. The filtered reads were mapped to the sequence of pLM1686 plasmid using BWA v0.7.17, when scaffolds in some isolates only aligned to partial of pLM1686.

**Data availability.** The 60 *L. monocytogenes* genome assemblies sequenced in this study were deposited at DDBJ/ENA/GenBank under the BioProject accession no. PRJNA795564. The other 23 genome assemblies sequenced in our previous study were deposited at DDBJ/ENA/GenBank under the BioProject accession no PRJNA447903. The genome accession numbers for each isolate used in this study were listed in Table S4 in the supplemental material.

## SUPPLEMENTAL MATERIAL

Supplemental material is available online only.
**SUPPLEMENTAL FILE 1,** XLSX file, 0.01 MB.
**SUPPLEMENTAL FILE 2,** XLSX file, 0.01 MB.
**SUPPLEMENTAL FILE 3,** XLSX file, 0.01 MB.
**SUPPLEMENTAL FILE 4,** XLSX file, 0.01 MB.
**SUPPLEMENTAL FILE 5,** PDF file, 0.4 MB.

## ACKNOWLEDGMENTS

This work was supported by grants from the National Natural Science Foundation of China (31800004) and National Key Research and Development Program of China (2018YFC1603800).

We declare that we have no known competing financial interests or personal relationships that have or could be perceived to have influenced the work reported in this article.

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
