## [Reviewer comments · Microbiology Spectrum]

Microbiology Spectrum

Dissecting *Listeria monocytogenes* persistent contamination in a retail market using whole-genome sequencing

Yan Wang, Lijuan Luo, Shunshi Ji, Qun Li, Hong Wang, Zhengdong Zhang, Pan Mao, Hui Sun, Lingling Li, Yiqian Wang, Jianguo Xu, Ruiting Lan, and Changyun Ye

Corresponding Author(s): Yan Wang, State Key Laboratory of Infectious Disease Prevention and Control, National Institute for Communicable Disease Control and Prevention, Chinese Center for Disease Control and Prevention.

Review Timeline:

Submission Date:	January 17, 2022
Editorial Decision:	February 18, 2022
Revision Received:	March 31, 2022
Editorial Decision:	April 15, 2022
Revision Received:	April 17, 2022
Accepted:	April 22, 2022

Editor: Luxin Wang

Reviewer(s): The reviewers have opted to remain anonymous.

Transaction Report:

DOI: <https://doi.org/10.1128/spectrum.00185-22>

February 18, 2022

Dr. Yan Wang

State Key Laboratory of Infectious Disease Prevention and Control, National Institute for Communicable Disease Control and Prevention, Chinese Center for Disease Control and Prevention.

Beijing

China

Re: Spectrum00185-22 (**Dissecting *Listeria monocytogenes* persistence and transmission in a retail market using whole-genome sequencing**)

Dear Dr. Yan Wang:

Link Not Available

Sincerely,

Luxin Wang

Journals Department
Reviewer comments:

Reviewer #1 (Comments for the Author):

The paper describes a study of CC87 strains of *Listeria monocytogenes* isolated from several retail markets during 2014 - 2015. This clone is a dominant one in listeriosis in China, and its presence in the environment and on foods is important. 83 isolates were characterized by wgMLST and 53 different wg-MLST types and 2 major clusters were described. Cluster A was isolated from several meat market stalls in the same retail market over a 6-month period. Cluster B isolates came from the same market, but was associated with aquatic product stalls.

1. Line 105: Are the cad genes required in host environments or more likely outside of a host? Please check this sentence for accuracy.
2. Line 431: The web address does not work.
3. Lines 225 - 227: Here the authors describe persistent and not persistent isolates. This makes sense based on the numbers of isolates, however, in this paper I don't get a sense of how expansive the survey was originally. They focus on CC87 here - were these the most isolated subtypes or selected because of their cause of more listeriosis in humans? How often were the markets sampled?
4. Lines 290 - 297: It's unclear here what the justification is for the conclusion that Cluster A could be upstream of a common source - is this because it was detected in 2 markets? What was the time frame of Cluster A detection in M1?
5. Line 303-304: Was Cluster A detected in the meat grinder?

Reviewer #2 (Comments for the Author):

The manuscript presents a descriptive analysis of *L. monocytogenes* isolates from retail markets using WGS. I have the following comments:

1. The authors may have got confused with some concept in 7-gene MLST and wgMLST. Specifically, singleton is a special clone that is clearly defined by 7-gene MLST, but the authors used singleton to describe taxa analyzed by WGS.
2. The authors did not explain their methodology clearly. For example, the authors divided CC87 into 12 types. wgST1, sgST2,... How was this type determined? How was the type used to detect persistence? Was it identified based on some threshold SNP/allele differences? If so, how were threshold determined?
3. How can we tell the incidence was due to persistence or repeated introduction of the same strain? The title of the manuscript contains transmission, but the data and analyses did not demonstrate transmission events.
4. Line 354, have the authors performed analysis to determine if some of the genes in phages may enhance *Listeria* adaptation?
5. Plasmids in *Listeria* often carry stress resistance genes, so identifying a plasmid may not explain the persistence/repeated contamination of CC87.
6. It is very difficult to read the allele differences in Figure 1. Figure 3 is not informative. What can we learn from figure 3?

Reviewer #3 (Comments for the Author):

The authors present an in-depth genomic assessment of a particular sequence type of *Listeria monocytogenes* found to be persistent over time in a market. The analyses are presented in a clear manner and the paper is easy to follow. The sequence type of *L. monocytogenes* has not been studied as much as other sequence types commonly found in other countries, so this work provides a useful set of data for further studies regarding this group. It would be useful to know how frequently ST87 is seen in cases of human illness to provide more context to the issue of strains of this ST being persistent.

Staff Comments:

Preparing Revision Guidelines

Please return the manuscript within 60 days; if you cannot complete the modification within this time period, please contact me. If you do not wish to modify the manuscript and prefer to submit it to another journal, please notify me of your decision immediately so that the manuscript may be formally withdrawn from consideration by Microbiology Spectrum.

Response to reviewers:

Reviewer comments:

Reviewer #1 (Comments for the Author):

The paper describes a study of CC87 strains of *Listeria monocytogenes* isolated from several retail markets during 2014 - 2015. This clone is a dominant one in listeriosis in China, and its presence in the environment and on foods is important. 83 isolates were characterized by wgMLST and 53 different wg-MLST types and 2 major clusters were described. Cluster A was isolated from several meat market stalls in the same retail market over a 6-month period. Cluster B isolates came from the same market, but was associated with aquatic product stalls.

1. Line 105: Are the cad genes required in host environments or more likely outside of a host? Please check this sentence for accuracy.

Response: we meant host of the plasmid which is the bacterium, not human host. We have corrected the sentence as "...which were related to adaptation to harsh environments" .

2. Line 431: The web address does not work.

Response: The web address was corrected as "<https://bigsddb.pasteur.fr/listeria/>"

3. Lines 225 - 227: Here the authors describe persistent and not persistent isolates. This makes sense based on the numbers of isolates, however, in this paper I don't get a sense of how expansive the survey was originally. They focus on CC87 here - were these the most isolated subtypes or selected because of their cause of more listeriosis in humans? How often were the markets sampled?

Response: A 12-month monthly survey was conducted from six retail-markets in Zigong, a town of Sichuan province in China in 2015. The isolates and survey results of pork retailing stalls was published as cited (Luo 2016). However, the aquatic stalls results were not published. We have provided the details in supplementary material. A total of 63 isolates from the aquatic stalls in the M1 market were obtained over the 12 months. By serotype 52 were 1/2b, by ST, 10 ST87 and 41 ST1166 (both STs belonging to CC87 as described in main text). By PFGE, 47 isolates were PT30, 40 of which were selected for this study.

Since the materials and methods section was at the end of the paper, we have also moved part of the strain selection to results so that reader can follow the paper without reading the methods first.

See Line 119-139 (the Line numbers matched to the Manuscript clean version, same as below):

"Selection of CC87 *Listeria monocytogenes* isolates for whole genome sequencing

In our previous study of surveillance of retail markets, we identified persistent contamination by the same isolates through PFGE analysis. Here, based on PFGE profiles, we selected part of CC87 *L. monocytogenes* isolates for genomic analysis to further confirm and investigate the persistent contamination that occurred in a retail market (M1 market). Twenty PT-30 isolates associated with pork products and

environments had been already published in Luo *et al.*'s study (21). Additionally, 43 isolates associated with aquatic products, other meat products (except for pork and aquatic products) and environments with PT-30 from the same market were used in this study. The surveillance of the aquatic products, other meat products and environments was done at the same time as the study of Luo *et al.* and isolates obtained were typed similarly. Except for the isolates from pork stalls, there were 78 isolates obtained in M1 market over the 12-month survey. By serotype 57 were 1/2b, by ST, 13 ST87 and 40 ST1166 (both STs belonging to CC87). By PFGE, 50 isolates were PT30, 43 of which were selected for this study. The distribution of the isolates among the aquatic stalls and other meat stalls over the sampling month were presented in Fig S1. For comparison, we chose two groups of isolates from the same surveillance: 1) nine PT30 *L. monocytogenes* isolates from other markets (M2 - M8), 2) 11 non-PT30 (PT27, PT315, PT317, PT331, PT322 and PT324) but CC87 isolates from all markets. The background information of each *L. monocytogenes* isolate was listed in Table 1. All the non-M1 markets were named from M2 to M8 randomly. Sixty isolates were subjected to sequence in this study and the other 23 isolates had been sequenced in our previous study (18). The complete genome of isolate ICDC-LM188 (accession No. CP015593), belonging to CC87/ST87, was used as a reference."

From both market surveys and human infections (ref 14 and ref 8), CC87 *L. monocytogenes* is prevalent in a variety of food products and environments in China and a major ST causing human infection. Therefore, the study focused on CC87. See Line 97-100: "ST87 *L. monocytogenes* is commonly isolated from food products, natural environments and sporadic listeriosis in China (8, 14, 15). Moreover, this subgroup of *L. monocytogenes* has been gradually recognized as most commonly causing listeriosis in China with the prevalence rate up to 34% (8, 16, 17)."

4. Lines 290 - 297: It's unclear here what the justification is for the conclusion that Cluster A could be upstream of a common source - is this because it was detected in 2 markets? What was the time frame of Cluster A detection in M1?

Response: Yes, based on the fact that one Cluster A isolate (ICDC_LM1201) was from M2, we think it was more likely that a common source existed in the upstream of the retailing chain. The Cluster A isolates were detected in M1 from February to July and October in 2015 while the M2 market isolate was isolated a few months earlier in November 2014. It is possible but less likely that Cluster A spread from M2 to M1 as Cluster A was only sampled once in M2.

The time frame of Cluster A isolates' detection in M1 was stated in Line 166-167 as "All Cluster A *L. monocytogenes* isolates were isolated from the same retail market M1 during February to July and October 2015 (Table 1),...".

5. Line 303-304: Was Cluster A detected in the meat grinder?

Response: Yes. 12 of 22 Cluster A isolates were detected in the meat stall (M1-S77). Six of these 12 isolates were detected in the meat grinder, which were separately isolated over six months (from February to July). We have added this in the MS. See Line 167-171: "More than half (12/22) of Cluster A isolates were repeatedly isolated from the same stall (M1-S77) over a 6-month period (February to July, 2015) in a monthly sampling (Fig. 2). Moreover, six of these 12 isolates were detected in the meat grinder of the M1-S77 stall, which were separately isolated in a six month period."

Reviewer #2 (Comments for the Author):

The manuscript presents a descriptive analysis of *L. monocytogenes* isolates from retail markets using WGS. I have the following comments:

1. The authors may have got confused with some concept in 7-gene MLST and wgMLST. Specifically, singleton is a special clone that is clearly defined by 7-gene MLST, but the authors used singleton to describe taxa analyzed by WGS.

Response: We referred any wgSTs not in the clusters as singletons as a generic term. To avoid any confusion, we have changed our description of them as unclustered wgSTs. See Line 159 and 162.

2. The authors did not explain their methodology clearly. For example, the authors divided CC87 into 12 types. wgST1, sgST2,... How was this type determined? How was the type used to detect persistence? Was it identified based on some threshold SNP/allele differences? If so, how were threshold determined?

Response: wgSTs were defined in the materials and methods section (see Line 508-510). Unfortunately, that section is at the end of the paper. We tried to briefly explain this in results so readers can understand the terminology and methodology. See Line 147-149: "Note that we did not attempt to develop a CC87 wgMLST scheme and were only using this approach to obtain best resolution for typing of the isolates rather than using the species specific core genome MLST scheme." In simplest terms, wgST is a sequence type based on the core genome of CC87 (not the core genome of the species).

Clusters were defined based on a cutoff of 4 alleles. Using Silhouette index, the number of clusters was 2 and the optimal cutoff was 4. This was added in revised MS in Line 159-160 as "Clusters were defined based on a cutoff of 4 alleles using Silhouette index."

3. How can we tell the incidence was due to persistence or repeated introduction of the same strain? The title of the manuscript contains transmission, but the data and analyses did not demonstrate transmission events.

Response: We have changed the title to persistent contamination. Apologies for the loose use of the word transmission. We referred all such events as contamination events rather than transmission as it is very difficult to confirm that there were transmission events.

It is difficult to differentiate between persistence and repeated introduction. We did not collect samples from sources of upstream of retail chain such as wholesale or pork processing plants for M1 pork products. Assuming all markets were sourcing products from the same wholesale chain, we would expect similar patterns of strain diversity as a result of repeated introduction. This was not the case for the pork retail markets based on the PFGE data. Our WGS data based on wgSTs or wgST clusters provided further evidence that it is more likely due to persistence rather than repeated introduction from another source.

4. Line 354, have the authors performed analysis to determine if some of the genes in phages may enhance *Listeria* adaptation?

Response: No, we haven't done any. It is beyond the scope of this study and will be future studies.

5. Plasmids in *Listeria* often carry stress resistance genes, so identifying a plasmid may not explain the persistence/repeated contamination of CC87.

Response: We agree that carriage of a plasmid doesn't mean the plasmid plays a role in adaptation. However, in this case, the genes carried by the plasmid or the plasmid have been shown to play a role in stress resistance and adaptation (Mao *et al.* study, ref 18). Therefore, it is reasonable for us to make an inference that this plasmid may play a role in the persistence of the plasmid-carrying strains in the environment.

6. It is very difficult to read the allele differences in Figure 1. Figure 3 is not informative. What can we learn from figure 3?

Response: We modified figure 1 so the number of allele differences is clearly legible on the figure.

Figure 3 showed how many genes are shared among the different phage types. Given that we didn't detail these in the text, we have moved figure 3 to the Supplemental Materials as Fig S2.

Reviewer #3 (Comments for the Author):

The authors present an in-depth genomic assessment of a particular sequence type of *Listeria monocytogenes* found to be persistent over time in a market. The analyses are presented in a clear manner and the paper is easy to follow. The sequence type of *L. monocytogenes* has not been studied as much as other sequence types commonly found in other countries, so this work provides a useful set of data for further studies regarding this group. It would be useful to know how frequently ST87 is seen in cases of human illness to provide more context to the issue of strains of this ST being persistent.

Response: The latest data of human listeriosis prevalence of ST87 *L. monocytogenes* in the introduction was added as "ST87 has also been reported to be the most common *L. monocytogenes* ST causing listeriosis in China with prevalence rate up to 34%." See Line 97-100: "ST87 *L. monocytogenes* is commonly isolated from food products, natural environments and sporadic listeriosis in China (8, 14, 15). Moreover, this sub-group of *L. monocytogenes* has been gradually recognized as most commonly causing listeriosis in China with the prevalence rate up to 34% (8, 16, 17)."

April 15, 2022

Dr. Yan Wang

State Key Laboratory of Infectious Disease Prevention and Control, National Institute for Communicable Disease Control and Prevention, Chinese Center for Disease Control and Prevention.

Beijing

China

Re: Spectrum00185-22R1 (Dissecting *Listeria monocytogenes* persistent contamination in a retail market using whole-genome sequencing)

Dear Dr. Yan Wang:

Link Not Available

Sincerely,

Luxin Wang

Journals Department
Reviewer comments:

Reviewer #3 (Comments for the Author):

The authors' modifications have improved the clarity of the manuscript.

1. on lines 123 and 126, it is unclear what 'PT-30' refers to.
2. in figure 2, it would be useful to make the lines of the tree thicker to see them more clearly.

Staff Comments:

Preparing Revision Guidelines

Please return the manuscript within 60 days; if you cannot complete the modification within this time period, please contact me. If you do not wish to modify the manuscript and prefer to submit it to another journal, please notify me of your decision immediately so that the manuscript may be formally withdrawn from consideration by Microbiology Spectrum.

Response to reviewers:

Reviewer comments:

Reviewer #3 (Comments for the Author):

The authors' modifications have improved the clarity of the manuscript.

1. on lines 123 and 126, it is unclear what 'PT-30' refers to.

Response: PT is short for pulsotype which is identified by PFGE. "PT-30" was replaced by "Pulsotype-30" when it first appear in the MS. The sentence was changed to "Twenty Pulsotype-30 (PT-30) isolates associated with pork products and environments had been already published in Luo et al.'s study."

2. in figure 2, it would be useful to make the lines of the tree thicker to see them more clearly.

Response: The lines of the tree have been made thicker in Figure 2.

April 22, 2022

Dr. Yan Wang

State Key Laboratory of Infectious Disease Prevention and Control, National Institute for Communicable Disease Control and Prevention, Chinese Center for Disease Control and Prevention.

Beijing

China

Re: Spectrum00185-22R2 (Dissecting *Listeria monocytogenes* persistent contamination in a retail market using whole-genome sequencing)

Dear Dr. Yan Wang:

Your manuscript has been accepted, and I am forwarding it to the ASM Journals Department for publication. You will be notified when your proofs are ready to be viewed.

Sincerely,

Luxin Wang

Journals Department
Supplemental file 2: Accept

Supplemental file 4: Accept

Supplemental file 3: Accept

Supplemental file 1: Accept

Supplemental file 5: Accept